# A Sensor System for Detecting and Localizing Partial Discharges in Power Transformers with Improved Immunity to Interferences

**DOI:** 10.3390/s19040923

**Published:** 2019-02-22

**Authors:** Petr Drexler, Martin Čáp, Pavel Fiala, Miloslav Steinbauer, Radim Kadlec, Miloš Kaška, Lubomír Kočiš

**Affiliations:** 1Department of Theoretical and Experimental Electrical Engineering, Faculty of Electrical Engineering and Communication, Brno University of Technology, 616 00 Brno, Czech Republic; capm@feec.vutbr.cz (M.Č.); fialap@feec.vutbr.cz (P.F.); steinbau@feec.vutbr.cz (M.S.); kadlec@feec.vutbr.cz (R.K.); 2TES s.r.o., 674 01 Třebíč, Czech Republic; kaska@tes.eu; 3EGU - HV Laboratory a.s., 190 11 Prague, Czech Republic; kocis@egu-vvn.cz

**Keywords:** partial discharge, ultra-high frequency, spatial localization, electromagnetic interference

## Abstract

The paper reports on the solution, principles, and application results related to a system for diagnosing main transformers in power plants via the radiofrequency method. The subject of the diagnostics is the occurrence of partial discharge activity in transformers. The technical solution of the system is characterized in the introductory section of the article. There then follows a description of the operating principle and the implemented novel advanced methods for signal detection and source localization. The results obtained from practical application of the system within the diagnostics of high-power transformers are presented as well. Because ambient electromagnetic disturbance was recognized as a major issue during the system development, novel detection methods were proposed, implemented, and verified. The principal approach utilizes an external radiofrequency sensor to detect outer impulse disturbance and to eliminate disturbance-triggered acquisitions, and it also ensures direct real-time visualization of the desired impulse signals. The ability of weak signal detection was verified via artificial impulse signal injection into the transformer. The developed detection methods were completed with localization techniques for signal source estimation. The desired impulse signal was detected and localized during full operation of the main transformer, despite the presence of strong electromagnetic interference.

## 1. Introduction

Reliable production and distribution of electrical energy are important preconditions for sustainable development within industry and, generally, human society. At present, the energy is produced mainly at central facilities (power plants), where it is also transformed and subsequently transported to the consumer. The production and distribution chain comprises various blocks with specific functions, and the safety and reliability of each of the blocks is intensively monitored.

One of the key components within the chain is a main transformer (generator transformer) in a power plant. This device transforms the voltage level on the output of a turbine-driven generator (typically around 20 kV) to a level suitable for long-distance electricity transmission (typically around 400 kV).

The transformer carries significant electrical power ranging from tens to hundreds of megawatts, and it thus assumes considerable dimensions (Figure 1). Since the voltage level handled by a main transformer can reach up to hundreds of kilovolts, the resulting high electric field intensity places strict demands on the working insulation, which is a part of the transformer structure. Another spurious effect faced by the insulation is the relatively high internal temperature caused by the high-power transmission and associated power loss. Combined paper-oil winding insulation is frequently used in such transformers.

The insulation installation embodies a demanding portion of the transformer assemblage procedure. The insulation has to withstand long operation under high electric field intensities. As working insulation plays an important role in a transformer’s reliability and safety, its condition needs to be observed in order to prevent the development of serious partial discharge (PD) activity. The partial discharge is the effect where an electrical discharge bridges over only a part of the insulation between electrodes with different potentials and a certain amount of an electric charge is transported [1]. In the above-mentioned high power main transformers, PDs are typically located in small cavities inside the paper-oil insulation; at such spots, the dielectric strength is reduced. Partial discharges can occur in these cavities, usually at certain phase instants of the harmonic voltage on the winding [2]. Long-term exposure of the cavity walls to a partial discharge causes unwanted chemical and structural decomposition of the insulation. Any progressive insulation damage resulting from PD activity weakens the insulating ability. Such a deterioration process then generates gas, typically hydrogen. At a certain stage of the insulation damage, an intensive arc may develop, leading to a catastrophic transformer failure.

The outer manifestations of PD activity commonly consist in voltage transients on the power lines of an electrical device. Quantitative characterization of the PD is impeded by the fact that it occurs inside devices and is not directly accessible for observation and magnitude measurement. Thus, magnitude quantification is often expressed by the quantity referred to as apparent charge. This value embodies the amount of a charge in Coulombs which, when injected into the device, causes the same magnitude of voltage transients as the PD itself. The PD apparent charge in a high-power transformer typically ranges between 10^1^ pC and 10^6^ pC.

Due to the seriousness of potential consequences, PD activity is intensively monitored. Various detection and diagnostic methods have been developed over time. The detection techniques exploit relevant side effects, including chemical decomposition of the medium; electrical transients in the power lines; and emission of acoustic signals and radiation in the radiofrequency and optical regions of the electromagnetic spectrum.

One of the central, classic approaches lies in chemical analysis of PD-generated gases dissolved in transformer oil [3]. The dissolved gases analysis (DGA) allows us to reliably determine the oil composition and, consequently, the condition of the insulation. Nevertheless, this technique does not facilitate instantaneous PD activity monitoring and cannot prevent a failure in cases of rapid insulation degradation. Another approach is the well-known electrical method, where the transient signals generated by rapid charge transport are detected [1,4]. The transients are superposed on the harmonic voltage of the transformer’s power lines. This technique utilizes coupling transducers to sense the transients on the high voltage potential. Because of the transients’ propagation along the power lines and the related high-frequency attenuation, the frequency content of the detected signal lies in the lower frequency domain (medium and high frequencies). The electrical method allows on-site, real-time monitoring of PD activity; however, it may exhibit difficulties in recognizing the internally PD-generated signal and false disturbing signals from other sources. Another eminent, intensively developed method is the acoustic approach, which detects the sound emitted by the PD, mainly at ultrasonic frequencies [5]. The crucial advantage consists in the possibility of installing the sensors outside the transformer vessel; further, estimation of the spatial position of the PD via multiple sensors and localization methods is feasible [6]. However, the acoustic method offers only rather low sensitivity and, as such, used to be combined with, e.g., the electrical technique for periodic triggering of the sound signal acquisition [7] and the consequent averaging. Another issue affecting the acoustic method consists in multipath acoustic signal propagation, with different velocities impeding the source localization. The set of recent approaches includes also the optical method, which senses light radiation from a discharge by means of optical detectors [8] and fibers [9].

The latest trend in the field is to prefer detection methods that, in addition to online PD detection, ensure also relevant localization. An intensively studied technique rests in the ultra-high frequency procedure for detecting and localizing partial discharges (PD UHF), which provides some unique advantages, as described below [10]. Interestingly, this approach has recently been combined with other classic methods in order to improve the diagnostic yield [11]. Another advantageous approach, which utilizes the UHF detection, is so-called “gating” or “noise gating” technique. This approach exploits the PD-generated UHF signal to trigger the PD signal acquisition measured via another measurement technique. Recently published experiments were using the UHF gating technique in combination with electrical method according to the standard IEC60270 [12].

Despite its exclusive features, the UHF method can nevertheless suffer from electromagnetic interference (EMI) when used in a real environment; therefore, significant effort has been made to investigate shielding effects in transformer vessels [13]. A novel approach to overcome this limitation is presented within our paper.

## 2. PD UHF Detection Method

The introduction and evolution of the PD UHF method were allowed by the current rapid progress in high-frequency and high-speed electronics and instrumentation. The technique detects PDs via sensing their electromagnetic emissions. The main spectral content of a PD-radiated electromagnetic (EM) signal lies between hundreds of MHz and units of GHz (UHF band) [14]. Since the PD is a short transient phenomenon, the emitted radiation has impulse-like character, with a transient’s duration reaching the sub-nanosecond domain.

EM waves propagate inside the transformer vessel with relatively low attenuation in cases of a free path in the oil medium; however, a wave can propagate at a certain (but somewhat higher) attenuation also through the transformer winding [15]. Thus, it is also possible to detect PDs that originate inside the winding structure.

A typical technical solution utilizing the PD UHF method relies on a sensor unit or a set of these; high-frequency circuits for signal conditioning; and a recording device and a computer for signal processing and PD events evaluation (Figure 2). All steps of the UHF PD diagnostic procedure were recently intensively examined. In this context, we can mention studies discussing the signal propagation mechanisms inside the metal transformer vessel with a complex inner structure, its influence on the attenuation and multipath propagation, and the impact on the signal source localization [16]. Significant effort was made within the design of UHF sensors of impulse broadband signals and optimal design of the high-frequency signal processing chain [17,18,19]. Research was conducted also in the calibration of the PD UHF detection chain [20].

The signals at a sensor output are recorded and stored or directly passed on for follow-up processing. An important consequent step consists in signal analysis to detect the PD-related impulse. In real-field employment of the UHF method, it is often necessary to distinguish between a PD impulse signal and a possible interfering signal [21]. Such a decision procedure then embodies one of the central aspects and benefits of the work presented herein and will be described in a dedicated chapter.

An advantageous capability of the PD UHF method lies in estimation of the PD source spatial location in the transformer vessel by means of triangulation. Such a procedure exploits mutual time relations of the acquired signals. Since the exact PD occurrence time is unknown, the time of arrival (TOA) parameters of the signals cannot be evaluated; instead, the time difference of arrival (TDOA) parameters can be calculated and consequently used for the localization. The TDOA represents the time intervals between particular impulse signal detections in each channel of the recording device. Generally, for the three-dimensional (3D) TDOA spatial localization, four suitably positioned sensors are required [22]. Further, the calculation of the PD source space coordinates exploits the solution of a set of hyperbolic equations. In addition to the TDOA-based localization, other approaches have been developed; some of them evaluate the received signal strength [23,24]. The advantage consists in the fact that the techniques do not rely on accurate timing of the signal arrival determination.

Several methods are available for the estimation of TDOA parameters. Because of the high propagation velocity of the EM wave in the medium and the relatively short distances, it is essential to estimate the parameters with sub-nanosecond precision. The classic approach determines the time of threshold crossing by the PD signal; this procedure is not immune to noise added to the signal. The method exploiting the energy accumulation curve (or its modifications) can yield better results in this respect [25,26]. Other approaches utilize, e.g., analysis of the cross-correlation function of two waveforms containing PD signals [25] or the statistically supported approach [27], and they can also improve the TDOA estimation accuracy. However, the result may exhibit sensitivity to the distortion and mutual similarity of the signals.

One of the often mentioned benefits of the PD UHF detection technique is the potentially high sensitivity, the reason being that the UHF signal detection is performed in the metal transformer vessel. The vessel should provide sufficient shielding against external electromagnetic disturbance of various origins. Such a view is based on two presumptions: First, the transformer vessel provides sufficient shielding and does not contain weak points that allow interference penetration; second, interference sources are absent from the inside of the vessel itself and connected armatures. Both of the presumptions, however, are difficult to fulfil completely in a real case. During our research and development of the PD UHF sensors system for transformer diagnostics, we registered violations of the two conditions specified above. Such adverse state then did not allow us to detect weak PD simulating signals. In order to overcome this issue, we designed, implemented, and verified a novel solution for eliminating the disturbance impact on weak impulse signal detection. The approach exploits full signal waveform acquisition as well as processing and visualization combined with discrimination of acquisitions disturbed by the interference. The developed detection method utilizes localization techniques for signal source estimation, all adopted for specific configuration of the sensor system.

## 3. PD UHF Diagnostic Sensor System

The development of the PD UHF diagnostic system was initiated by the need of periodic long-term operational monitoring of high-power transformers with respect to partial discharge activity. Another incentive was to introduce a tool facilitating research into innovations within PD UHF detection, PD signal analysis, and PD localization.

The target devices for the diagnostics consisted in eight three-phase main transformers installed in the Dukovany nuclear power plant, the Czech Republic (Figure 1b). The units are fed by turbine generators and ensure the voltage transformation of 16/420 kV with the nominal power of 300 MVA. The dielectric filling is embodied in 51 cubic meters of transformer oil. The transformers had been installed instead of the original units during a major overhaul of the plant; therefore, it was possible to carry out structural measures allowing incorporation of PD UHF detection sensors. Such measures involve hollow Teflon inlets penetrating into the transformer vessel through its metal wall. Because of the constructional and safety restrictions, only four inlets could be put in a linear configuration on one side wall of the transformer vessel, as shown in Figure 3. Although such a spatial configuration of the sensors is far from the ideal arrangement, it proved satisfactory for diagnostic purposes.

Other problems of interest included limited length and inner diameter of the inlet. This aspect then constrained the design and dimensions of the antennas. An investigation into selected types of antennas was performed yield the most suitable option. One of the central demands consisted in securing a greater length to improve the sensitivity, mainly at lower frequencies; another requirement was a sufficient sensitivity in broader spectral range in order to be able to receive signals in the UHF band. The best compromise properties for the given restrictions were found in the monopole conical antenna shown in Figure 4, together with its dimensions and *s*_11_ parameter.

The overall setup of the PD UHF diagnostic system comprises the main unit, five sensor units, and a set of signal transmission and control cables. The sensor unit consists of the above-mentioned receiving antenna, a UHF signal processing chain, power supply circuits, and a shielding case. The signal processing chain was developed on the basis of laboratory experiments with the detection of low-level impulse signals. A simplified schematic of the sensor unit’s inner structure is presented in Figure 5. The receiving antenna is covered by a dielectric shell, and a signal limiter is connected at the antenna’s output to protect the following circuitry components against an excessive signal magnitude. Signals at lower frequencies originating from other discharge activities are suppressed by a high-pass filter, whose corner frequency is set to 100 MHz. For example, the reception of EM signals generated by a corona discharge is greatly suppressed, since their spectral content is very low in the UHF band [28]. The bandwidth of the sensor equals 3 GHz; however, the resulting maximum operating frequency of the system is given by the digitizer in the main unit and corresponds to 1 GHz. After filtering, the signal level is increased by a variable gain stage. The possible maximum signal gain range of 40 dB turned out to be sufficient for the processing of signals with an anticipated magnitude span. The power to the sensors’ active circuits is supplied by a bias-T block, which separates the DC power supply from the RF transmission line.

Each sensor unit is embedded in a semi-opened metal casing and connected to the inlet via an EMI gasket (round configuration of the finger contacts). The gasket prevents the EMI penetration into the transformer vessel through the junction between the sensor case and the inlet. The implementation of a sensor unit is represented in Figure 6.

Each of the assemblies is connected to the main unit via a pair of double-shielded coaxial cables facilitating the RF signal transmission, power supply, and transmission of the amplifier’s gain control signal.

The components of the main unit (Figure 7) are embedded in a mobile metal case which provides weather sheltering and ventilation, simultaneously with protection against EMI penetration.

The block scheme of the main unit is shown in Figure 8. The key component of the unit consists in a high-speed digitizer equipped with four analog input channels having the analog bandwidth of 1 GHz and the sampling rate of 2 GSps with 10-bit resolution. The digitizer is connected to the control computer via a PCIe bus to enable fast data transfer for the follow-up signal processing and visualization. Other important parts of the main unit are a remote power supply block to feed the sensor units via the signal transmission cables; a remote gain control block for setting the signal gain of the sensors; a net synchronization block to determine the PD occurrence in relation to the phase of the net voltage; and a main power supply block with EMI filtration. The main unit is equipped with the Ethernet interface, allowing remote operation during the diagnostic procedure. In this case, an external computer is connected via a shielded twist pair (STP) cable.

The connection between the main unit and each sensor setup is provided by a pair of 17 m long coaxial cables with double shielding to increase the EMI immunity. The cable pairs ensure transmission of the received UHF and gain control signals and also facilitate remote power supply of the sensors’ circuitry. Since the TDOA parameters of the acquired signals are computed after the impulse detection, great attention was paid to equal lengths of the UHF signal transmission cables. The differences in the lengths were minimized using the time domain reflectometry (TDR) technique during the fabrication of the cables. The signal propagation differences are below 10 ps, a value sufficient for the given application.

The system’s software controls the signal amplification, acquisition, processing, and visualization. It exploits a multithread code and is optimized for the given hardware specification of the control computer to allow maximal utilization of the system resources. The software performs continuous visual monitoring of the signal waveforms and visualizes their phases in a diagram with respect to the phase of the net voltage, and it also calculates the signal source coordinates to visualize them in a 3D projection of the transformer vessel. Figure 9 shows the graphical interface of the software together with the relevant magnitude-phase diagram, which displays the relative magnitude of the detected discharge events (the red dots) and their phase positions in relation to the phases of the net voltage components (the blue, green, and red arrows). Further, it is possible to display the waveforms of the detected signals in the second window tab, as shown in Figure 10a.

The triggering of the signal acquisitions is based on crossing the adjusted threshold by any of the signals in the four channels of the digitizer. The signal gain of the sensors is automatically adjusted in order to obtain the full range of the digitizer’s inputs. The acquired signals can be visualized in real time and stored. After a successful signal acquisition, the time differences of arrivals (TDOA parameters) of the impulse signals are estimated. To achieve this goal, the energy accumulation curve (EAC) is derived for each signal [22]. The EAC represents the rate of the instantaneous signal energy rise and can be used for estimating the impulse signal time of occurrence with respect to the trigger time. Subsequently, the TDOA parameters are estimated via thresholding the EACs. Optionally, a search for the time instant of the EAC derivative maximum (the steepest signal energy rise) can be performed.

The third tab window of the software interface serves for the spatial localization of the signal source (the discharge event location). This task is computed using multiple acquisitions (multiple sets of TDOA parameters) in order to statistically improve the accuracy.

Two localization methods are implemented in the relevant software module: one utilizing an analytical approach [22], with the derived relations employed to compute the signal source from the TDOA parameters, and the other based on a comparative procedure. The TDOA parameters of the acquired signals are compared to the pre-calculated TDOA parameters for hypothetical locations inside the transformer. A model visualization of source localization in the software is shown in Figure 10b, and a more detailed description of the implemented localization methods is outlined in the section below.

## 4. Implemented Localization Methods

Because of the specific layout of the sensors with respect to the transformer geometry, it is not possible to unambiguously determine the position of the source. The computed source positions are represented by points on a circle. The line which connects the sensor positions passes perpendicularly through the center of the circle. This ambiguity, however, is not a serious restriction factor in the diagnostic procedure. The computed and visualized circle crosses the corresponding winding, and the signal source location can be estimated via examining the intersection.

As mentioned above, two methods for the spatial localization of the signal source are implemented: an analytical method and a comparative technique. The analytical option uses relations for Cartesian coordinates derived from the set of hyperbolic equations formulated for the known sensor positions S_1_, S_2_, S_3_, and S_4_, as shown in Figure 11a.

The location calculation consists in computing a set of non-linear hyperbolic equations. Since the absolute time of PD occurrence is unknown, the TDOA parameters have to be used as the input variables in the following set:
(1)$$\begin{array}{c} {\left(x-x_{S1}\right)}^{2}+{\left(y-y_{S1}\right)}^{2}+{\left(z-z_{S1}\right)}^{2}={\left(v\cdot t_{0}\right)}^{2},\\ {\left(x-x_{S2}\right)}^{2}+{\left(y-y_{S2}\right)}^{2}+{\left(z-z_{S2}\right)}^{2}={\left(v\cdot \left(t_{0}+t_{12}\right)\right)}^{2},\\ {\left(x-x_{S3}\right)}^{2}+{\left(y-y_{S3}\right)}^{2}+{\left(z-z_{S3}\right)}^{2}={\left(v\cdot \left(t_{0}+t_{13}\right)\right)}^{2},\\ {\left(x-x_{S4}\right)}^{2}+{\left(y-y_{S4}\right)}^{2}+{\left(z-z_{S4}\right)}^{2}={\left(v\cdot \left(t_{0}+t_{14}\right)\right)}^{2}, \end{array}$$
where *x*, *y*, *z* represent the three unknown coordinates of the signal source; *x*_Si_, *y*_Si_, *z*_Si_ (*i* = 1,2,3,4) are the sensor coordinates; *v* denotes the propagation velocity of the signal; and *t*_12_, *t*_13_, *t*_14_ stand for the TDOA parameters of the acquired signals. Further, *t*_0_ (as the fourth unknown variable) is the time instant between the PD occurrence and the first signal arrival to the first sensor. Although the given spatial sensor (Figure 11b) configuration is not ideal (from the perspective of unambiguous source localization), it allowed us to reduce the equation set (1), greatly simplifying its analytical solution. Such a reduction leads to the set of three equations for 2D spatial localization. When the position of the first sensor is in the origin of the coordinate system, the equation set is reduced as
(2)$$\begin{array}{c} x^{2}+y^{2}=v^{2}t_{0}^{2}\\ x^{2}+{\left(y-y_{S2}\right)}^{2}=v^{2}{\left(t_{0}+t_{12}\right)}^{2}\\ x^{2}+{\left(y-y_{S3}\right)}^{2}=v^{2}{\left(t_{0}+t_{13}\right)}^{2}. \end{array}$$


The variable *x* is derived from the first equation of the set (1) and substituted into the second and third equations of the set (2)
(3)$$\begin{array}{c} v^{2}t_{0}^{2}-y^{2}+{\left(y-y_{S2}\right)}^{2}=v^{2}t_{0}^{2}+2v^{2}t_{0}t_{12}+v^{2}t_{12}^{2}\\ v^{2}t_{0}^{2}-y^{2}+{\left(y-y_{S3}\right)}^{2}=v^{2}t_{0}^{2}+2v^{2}t_{0}t_{13}+v^{2}t_{13}^{2} \end{array}$$


By further expansion of the quadratic terms, we get
(4)$$\begin{array}{c} -2yy_{S2}+y_{S2}^{2}=2v^{2}t_{0}t_{12}+v^{2}t_{12}^{2}\\ -2yy_{S3}+y_{S3}^{2}=2v^{2}t_{0}t_{13}+v^{2}t_{13}^{2} \end{array}$$


From the first equation of the set (4), we can derive the formula for *y* as
(5)$$y=\frac{y_{S2}^{2}-2v^{2}t_{0}t_{12}-v^{2}t_{12}^{2}}{2y_{S2}}=\frac{y_{S2}}{2}-\frac{v^{2}t_{0}t_{12}}{y_{S2}}-\frac{v^{2}t_{12}^{2}}{2y_{S2}}.$$


Introducing (5) into (3), we get
(6)$$t_{0}=\frac{v^{2}\left(y_{S2}t_{13}^{2}-y_{S3}t_{12}^{2}\right)+y_{S3}y_{S2}^{2}-y_{S2}y_{S3}^{2}}{2v^{2}\left(y_{S3}t_{12}-y_{S2}t_{13}\right)},$$
and, subsequently, the final relations for the *x*, *y* coordinates:
(7)$$\begin{array}{c} y=\frac{y_{S2}}{2}-\frac{v^{2}\left(y_{S2}t_{13}^{2}-y_{S3}t_{12}^{2}\right)+y_{S3}y_{S2}^{2}-y_{S2}y_{S3}^{2}}{2y_{S2}\left(y_{S3}t_{12}-y_{S2}t_{13}\right)}t_{12}-\frac{v^{2}t_{12}^{2}}{2y_{S2}},\\ x=\sqrt{v^{2}(\frac{v^{2}\left(y_{S2}t_{13}^{2}-y_{S3}t_{12}^{2}\right)+y_{S3}y_{S2}^{2}-y_{S2}y_{S3}^{2}}{2y_{S2}\left(y_{S3}t_{12}-y_{S2}t_{13}\right)})-(\frac{y_{S2}}{2}-\frac{v^{2}\left(y_{S2}t_{13}^{2}-y_{S3}t_{12}^{2}\right)+y_{S3}y_{S2}^{2}-y_{S2}y_{S3}^{2}}{2y_{S2}\left(y_{S3}t_{12}-y_{S2}t_{13}\right)}t_{12}-\frac{v^{2}t_{12}^{2}}{2y_{S2}})} \end{array}$$


The derivations of relations (1)–(7) are based on three sensors positions. Since we have four sensors in total, the source position is computed three times for the sensor combinations (S_1_-S_2_-S_3_; S_1_-S_2_-S_4_; S_1_-S_3_-S_4_). The resulting coordinates are obtained as the average values yielded from the three results for *x* and *y*. This approach improves the accuracy. The localization results are then visualized by means of either Bézier curves in the transformer side view or axonometric view. The analytical method allows very fast localization of the signal source; however, its accuracy is sensitive to that of TDOA parameters estimation. Because of this significant sensitivity, the outcomes are further filtered to remove the results showing the source position outside the vessel. An example of the analytical localization procedure is shown in Figure 12, where the source of the artificial impulse signal was installed in the third inlet.

With more consistent TDOA parameter estimation results, it is advantageous to use the comparative method, which combines the outcomes of the TDOA estimation for all signal acquisitions. The whole transformer volume is divided into a mesh of points with the spacing of 10 cm. Over this mesh, a four-dimensional matrix *M*(*x*,*y*,*z*,*t*) is pre-calculated. The matrix contains the TDOA parameters for each point according to the equation set:
(8)$$\begin{array}{c} t_{1}=\sqrt{{\left(x_{S1}-x\right)}^{2}+{\left(y_{S1}-y\right)}^{2}+{\left(z_{S1}-z\right)}^{2}}\\ t_{2}=\sqrt{{\left(x_{S2}-x\right)}^{2}+{\left(y_{S2}-y\right)}^{2}+{\left(z_{S2}-z\right)}^{2}}\\ t_{3}=\sqrt{{\left(x_{S3}-x\right)}^{2}+{\left(y_{S3}-y\right)}^{2}+{\left(z_{S3}-z\right)}^{2}}\\ t_{4}=\sqrt{{\left(x_{S4}-x\right)}^{2}+{\left(y_{S4}-y\right)}^{2}+{\left(z_{S4}-z\right)}^{2}}. \end{array}$$


For simplification, the TDOA parameters are calculated for direct propagation paths between the actual point and the sensors, and they are then given as:
(9)$$\begin{array}{c} t_{12}=t_{2}-t_{1}\\ t_{13}=t_{3}-t_{1}\\ t_{14}=t_{4}-t_{1}. \end{array}$$


Further, a parameter which defines the tolerance interval of comparison accuracy is introduced as the quantity with the unit of nanosecond. This parameter can be modified by the operator. The localization consists in comparing the currently estimated TDOAs from the signal acquisition with the matrix elements that contain pre-calculated TDOAs. If the result (the TDOA estimate) falls within the range given by the mesh point and the tolerance interval, the value of the target variable for the defined vicinity of the point is incremented. The defined vicinity for each point represents a cube with the side dimension of 50 cm, as shown in Figure 13a. Within the progressive localization for each acquired set of signals, the target variables in the matrix of results are successively incremented. Only locations with more than a half of the maximal value of the target variable are visualized. Using such filtering, locations with a repeated count of occurrence are preferred and selected. Using above described principle, the comparative method has reduced sensitivity to not very consistent set of TDOA estimates for a set of signals acquisitions. An example of comparative localization is shown in Figure 13b; here, the source of the artificial impulse signal was installed in the third inlet.

## 5. Electromagnetic Interference Issues in PD Detection

The functionality of the developed diagnostic sensor system was verified within experimental measurement on eight 300 MVA power transformers at the Dukovany power plant. The testing cycle was carried out under full operating conditions. The sensor units were installed in the inlets, and pertinent impulse signals on their outputs were sought. Since the transformers had been manufactured comparatively shortly before the tests, no observable PD-related discharge activity was expected; however, we observed intensive discharge activity in transformer 1. The recorded signals contained strong impulse-like waveforms. An example of the recorded signals is shown in Figure 14a; Figure 14b then displays the signal source localization results, indicating that the signal arrived from the aperture of the 22 kV feeding bushing.

Further investigation showed that the source of interference was situated in one of the high voltage bushings that feed the transformer from a turbo-generator; we then inferred that the source rests in discharge activity on the inner 22 kV line support. The bushing structure comprises a dilatation compensating junction (Figure 15a), which also allows exiting the signal out of the bushing. The recorded signals are strongly dispersed in time, due to the multiple path propagation of the signal from a distant source. The disturbance source was confirmed when one of the sensors had been removed from its vessel inlet and positioned close to the spot where the interfering signal exits the dilatation junction of the bushing.

Similar effects were observed in transformers 2 to 5; transformer 6, however, exhibited different behavior, with all of the internal sensors indicating comparable signal magnitudes. Examining the vicinity of the bushing junction by means of a commercial UHF probe (Figure 15b) did not yield the source of the disturbance inside the bushing; such an outcome then indicated that relevant signals from distant sources (outside the transformer’s armatures) could penetrate the vessel to hamper the detection of weak signals, including the PD-related ones.

Previously published papers often mention that the metal transformer vessels ensure convenient shielding to facilitate low-level PD UHF signal detection. However, when evaluating the results of the experimental measurement, we formulated a conclusion not proposed to the date, namely, that in certain types of transformers an outer disturbance can penetrate into the transformer’s vessel. This constitutes a serious problem since such a condition can impede successful detection of a weak PD electromagnetic signal. Further in this context, it is not always possible to provide convenient shielding, due to the high voltage safety and other structural measures.

## 6. Method for the Discrimination of Disturbed Acquisitions

In order to overcome unexpected disturbance occurrences, we designed and verified a new method for detecting weak signals generated by partial discharges: the discrimination of disturbed acquisitions (DDA) technique. This method was verified via detecting an artificial impulse signal injected into the transformer vessel.

As only new transformers with unmeasurable PD activity were available, we built an artificial impulse signal generator that simulates the PD electromagnetic radiation when connected to an antenna. The device utilizes rapid avalanche breakdown in the transistor structure and produces a negative polarity impulse with a frontal rise time below 300 ps and full width at the half of the maximum time of 700 ps. The peak value at the output equals 5.9 V. The impulse waveform is represented in Figure 16a. The artificial impulse should simulate the EM radiation of PDs in view of its waveform, time duration, and spectral composition. The magnitude spectrum of the impulse is shown in Figure 16b. The PD simulating generator was built into the same case as the PD sensors and comprised an identical conical antenna, allowing us to install the generator instead of one of the sensors into the inlet in the transformer’s vessel.

In the standard operating mode of the sensor system, the artificial impulse signals were not detected, the reason being the presence of strong impulse disturbance. The detection was nevertheless successful after the DDA method had been implemented. The diagnostic system was equipped with an external sensor designed for detecting the outer EM impulse disturbance, Figure 17a. The configuration of the external sensor resembled that of the internal sensors (Figure 5).

The DDA method utilizes the multi-triggering ability of the built-in digitizer. The signal acquisitions can be triggered via signal threshold-crossing when this occur in any of the input channels. The external sensor is connected to an input channel instead of an internal sensor. Thus, only 3 internal sensors are used for the transformer’s internal signal detection; such an arrangement, however, suffices for the signal source localization. The external sensor is installed preferably near a location where the disturbance can penetrate (or leak away from) the transformer’s vessel. A convenient option is the service footbridge on the top of the transformer (Figure 17b). If placed here, the external sensor is close to the bushings.

Subsequently, the signal acquisition procedure begins. The signal gains of the internal sensors are set to a suitable level. The disturbing signal from the bushing (or the outer source) propagates inside the vessel towards the internal RF sensors and also outside, where it is detected by the external sensor to trigger the signal acquisition process. A strong disturbing impulse signal may overlap a weak PD (or artificial) one in the internal sensor if they occur simultaneously; such a scenario, however, does not materialize regularly.

Subsequently the signal gain of the internal sensor is raised to such level when also weak signal (PD origin or artificial) trigger acquisitions. At this point we obtain two sets of signal acquisitions. One group of acquisitions are triggered by the disturbing signal by the signals from PD or artificial source. To pick the relevant acquisitions a sorting procedure is performed. The acquisitions triggered by the external sensor are discriminated from further data processing. These acquisitions were triggered by the disturbing signals. The rest are the acquisitions triggered by the low-level signals via internal sensors. The low-level signals are of internal origin and cannot penetrate to the external sensor with sufficient magnitude for triggering. These signals are further saved and processed for displaying or for computation of the source location.

## 7. DDA Method Demonstration

Using the implemented DDA method, the diagnostic sensor system can detect and display in real-time weak impulse signals generated in the transformer’s vessel under the presence of stronger impulse disturbance coming from other discharge activities outside the vessel. The demonstration of weak signal detection is presented below.

Figure 18 shows the experimental measurement setup, when all sensors are installed in the inlets. Such configuration is used within a standard detection mode. This mode doesn’t allow to acquire weak signals (as for example the internally generated UHF PD signals) in case of strong disturbance presence. In this case, stronger impulse disturbance signals will be recorded.

Figure 19 shows the waveforms of the disturbing signal acquired within the standard detection mode by means of four internal sensors. These waveforms are typically dispersed in time due to the complex propagation path to the sensors. Simultaneously, the high frequency components of the signal are attenuated, causing slow signal build-up in the frontal section.

In case of an external sensor configuration (utilized within the DDA mode), the external sensor is connected to the input of the third channel and installed close to the expected source of disturbance (the bushing dilatation junction). In order to demonstrate the performance of the DDA method, the above-described impulse generator was used for injecting an artificial impulse signal into the third inlet. The experimental setup is shown in Figure 20.

The acquired waveforms are shown in Figure 21. It is evident from the waveforms that (1) the EM signal arrived first to the external sensor on the third channel; (2) it exhibited the highest magnitude; (3) the signals in channels 1, 2, and 4 experienced significant dispersion. The first and second conditions are due to the external sensor having been positioned correctly, close to the source of disturbance in the bushing. The third point indicates that the disturbing signal propagated by a longer trajectory to the internal sensors (via the bushing). The frontal parts of the waveforms also rise only slowly because of strong attenuation of the high-frequency components of the impulse signal spectrum.

Within the DDA mode, the sensor gains are adjusted either by an autonomous algorithm or manually. In the case of proper gains setting, we can observe the waveforms of the amplified artificial impulse signal in channels 1, 2, and 4 (Figure 22). Channel 3 lacks a signal since it was used for the sensing and discrimination of the disturbance. The waveforms of the artificial signal apparently have a shorter time duration compared to the disturbing impulse signals; this is because they propagate only in the transformer vessel and do not experience significant complex multipath propagation and related distortion. The frontal part is also very steep, unlike that of the disturbing signal, indicating the presence of high-frequency components. It is also obvious from the vertical axes scale that their magnitude is significantly lower than that of the disturbing signal in Figure 19.

The acquired signals were consequently used for the experimental localization of the signal source. Figure 23 presents the source localization result obtained by means of the comparative method, showing that the source was correctly localized at the position of the third inlet, where the generator had been installed.

It has to be noted that the above-described measurement was conducted with the high voltage transformer under full operation. Very strong interference was present in the given environment. Despite such conditions, the weak EM impulse signal was successfully detected and localized.

In transformers exhibiting extremely low PD activity (typically a new device), the DDA method leads to a stoppage of the signal visualization process as there is no signal to satisfy the prerequisites for visualization.

## 8. Discrimination Based on Temporal Signal Dispersion

The system software has implemented another tool for discharge activity evaluation that can be alternatively used for disturbance discrimination. This tool utilizes the fact that impulse signals which are not generated internally in the vessel (disturbing signals) are characterized by significant temporal dispersion due to their complex propagation path from the bushings. Considering this, it is possible to separate signals with low temporal dispersion (desired internally generated signals) from the set of all acquisitions. To do so, adjustable limits that the signal cannot exceed in the sense of magnitude and time duration are introduced. These limits define a banned area—red rectangles in the waveform graphs in Figure 24. The temporal dimension of the banned area is adjustable via shifting its length by moving the division between the green and red stripes over the first waveform (marked by a green arrow in Figure 24). The magnitude dimension of the banned area (marked by a blue arrow in Figure 24) can be set as the percent from the channel range (marked by a yellow rectangle in Figure 24). In the case of waveforms shown in Figure 24, they will be excluded from further processing since their components leak in the banned area. Though, they are visualized for demonstration purpose.

Within the operation of this detection mode, the sensors’ amplifiers gain is set on high value in order to trigger the acquisitions by weak signals also. In this mode, the weak impulse signals (generated by PDs or artificially) can be detected without the need of external sensor. However, the disadvantage consists in the need of manual user tuning of the banned area limits to successfully separate the desired signals.

## 9. Conclusions

The PD UHF method embodies one of the most advanced approaches to the diagnostics of high voltage transformers as regards partial discharge activity monitoring. A significant advantage consists in the real-time detection and localization of partial discharges. Nevertheless, several precautions still have to be adopted. The electromagnetic signal radiated by the discharges is usually weak; thus, the effect of external disturbances must be considered. The field experiments clearly pointed to the fundamental importance of proper shielding as the PD signal level is considerably below the interference one. Common interpretations of the PD UHF method emphasize its excellent sensitivity arising from inherent shielding of the transformer’s vessel. In this context, however, we established that it is not sufficient to rely solely on the shielding properties, because in many cases sufficient additional shielding cannot be ensured due to various high voltage safety precautions and structural measures.

Importantly, most of the transformers examined by our research team exhibited discharge activity in the feeding bushings, and such a problem is difficult to resolve, as already proposed above. In one of the investigated devices we also observed the coupling of interference from a distant source, concluding that the penetration paths might be sought in the bushing dilatation joints and the 400 kV bushing on the extra high voltage side of the transformer.

To eliminate the deficiencies, we designed and verified a novel method for detecting the weak signals generated by partial discharges. The technique is combined with the real-time visualization of the acquired signals, allowing us to simultaneously discriminate the signals produced by spurious discharge activities and to observe the waveforms arriving from the transformer’s vessel. These signals can be of either PD or artificial origin (namely, injected to monitor the performance of the given detection method).

The Discrimination of Disturbed Acquisitions (DDA) technique was verified via detecting the weak artificial impulse signal injected into the transformer vessel. The DDA method utilizes the multi-triggering ability of the built-in digitizer as well as an external sensor to detect the outer disturbing signal. Within the DDA procedure, the acquisitions triggered by the external sensor are excluded from further processing as they have deteriorated through the action of the disturbance.

Further, we verified the technique that discriminates the temporal dispersed signals; this is an alternative approach to signal separation, based on setting such time and amplitude limits that the signal cannot cross over.

The verification of the DDA method proved the procedure’s capability of detecting weak artificial impulse signals and positively localizing their sources. Interestingly, the actual practical demonstration was carried out on transformers under full operation. Although interference was present in the testing environment, the weak artificial EM impulse signal was successfully detected and localized.

The developed technique was completed with localization methods to estimate the signal source, and these methods were adapted to the sensor system configuration.

The above-outlined principles, devices, and methods are implemented into the proposed diagnostic system, which finds application in periodic predictive diagnostic monitoring of main transformers in nuclear power plants.

At present, the literature offers discussions analyzing disturbance and its influence on partial discharge UHF detection. Our paper, however, predominantly outlines the particulars of the specific condition where the disturbance source is comprised directly in the transformer’s armatures and may thus be unresolvable via conventional approaches. A unique aspect of the technique consists in the possibility of ensuring real-time signal waveform observation together with fast signal source localization. To our knowledge, no previously published work describes a corresponding transformer diagnostic procedure employed in real conditions.

## Figures and Tables

**Figure 1 sensors-19-00923-f001:**
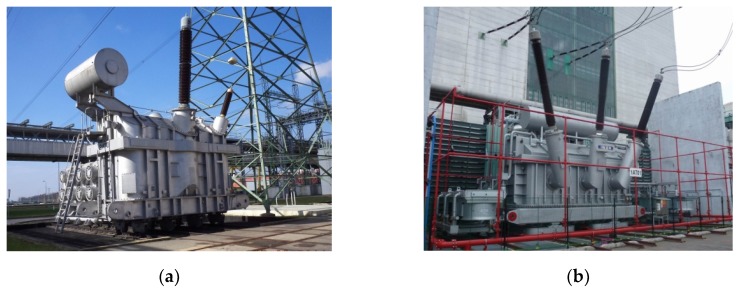
Examples of high-power main transformers: a single-phase 24/420 kV unit, 400 MVA (**a**); a three-phase 16/420 kV unit, 300 MVA (**b**).

**Figure 2 sensors-19-00923-f002:**
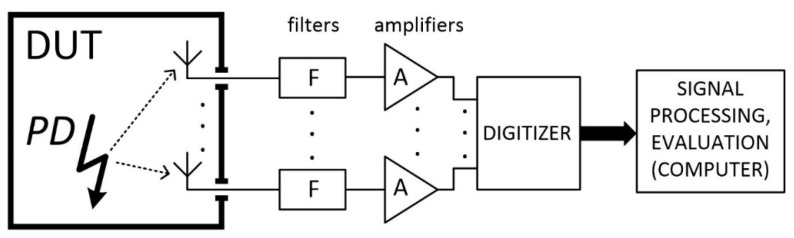
The basic setup for UHF detection of partial discharges.

**Figure 3 sensors-19-00923-f003:**
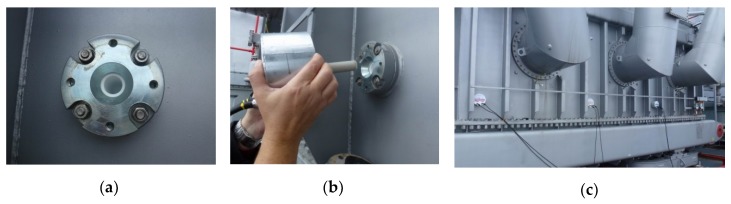
The antenna inlet on the side wall of the transformer (**a**); sensor installation (**b**); the installed sensors (**c**).

**Figure 4 sensors-19-00923-f004:**
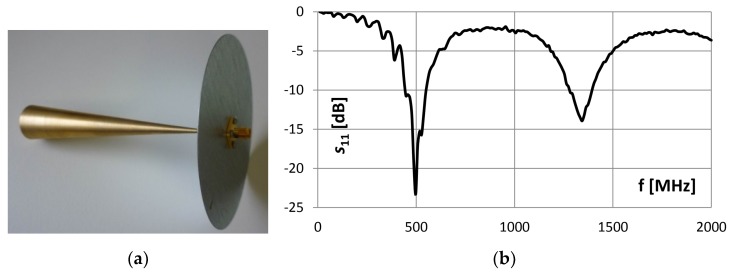
The monopole conical antenna for detecting PD radiated EM impulses (**a**), arm length 155 mm, end diameter of the arm 30 mm; reflection coefficient *s*_11_ of the monopole (**b**).

**Figure 5 sensors-19-00923-f005:**
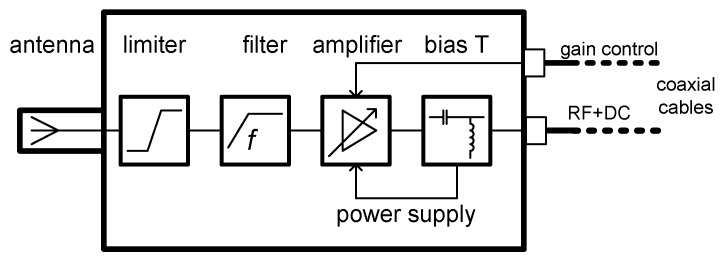
The principal schematic of the PD UHF sensor unit.

**Figure 6 sensors-19-00923-f006:**
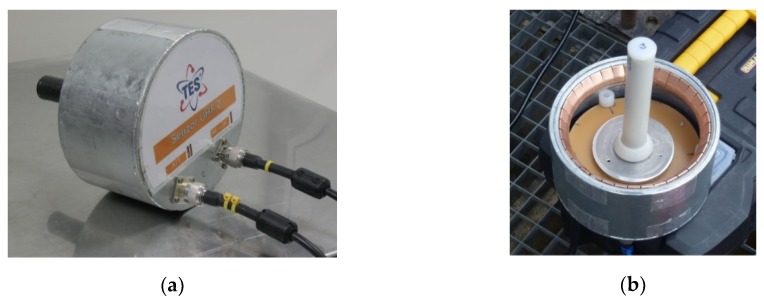
The PD UHF sensor unit: the rear (**a**) and front (**b**) sections.

**Figure 7 sensors-19-00923-f007:**
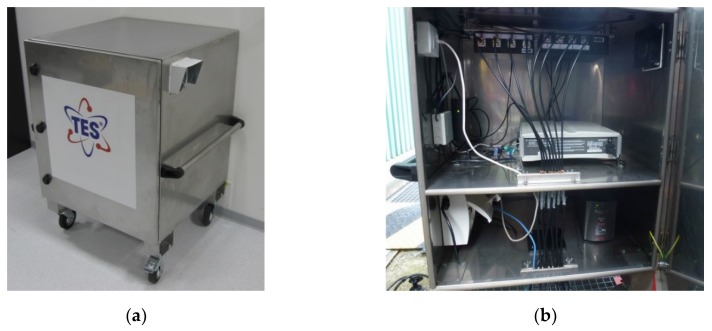
The main unit of the PD UHF diagnostic system: closed (**a**) and open (**b**) case views.

**Figure 8 sensors-19-00923-f008:**
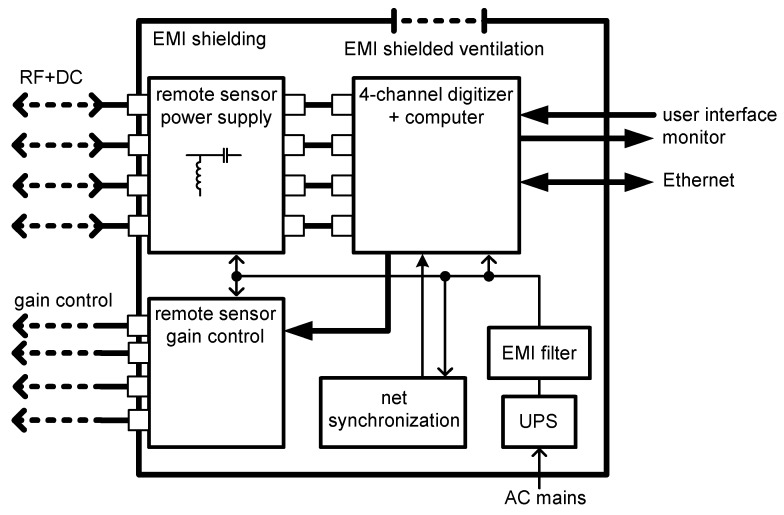
The principal block diagram of the main unit.

**Figure 9 sensors-19-00923-f009:**
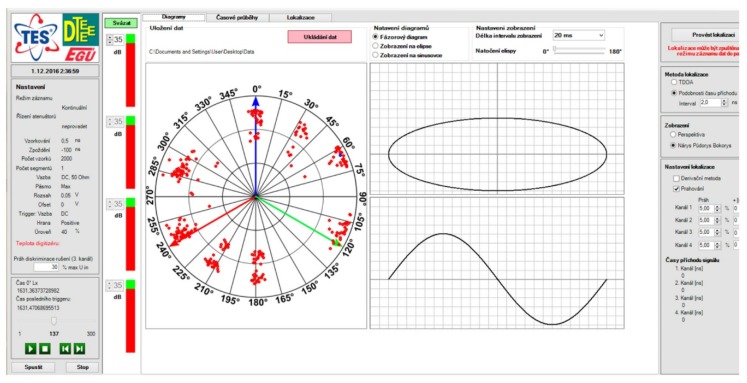
The general window of the PD diagnostic software, with impulse signal detection events (the red dots) in the magnitude-phase diagram.

**Figure 10 sensors-19-00923-f010:**
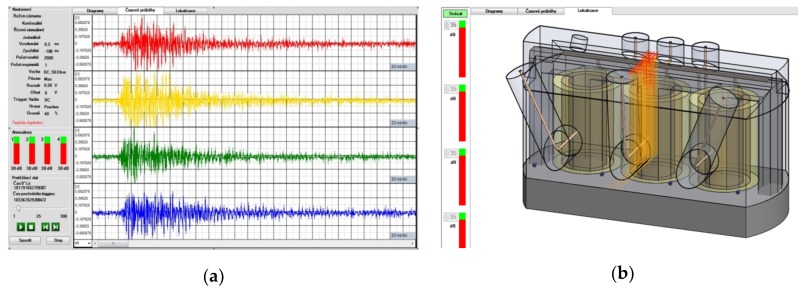
The software visualization of detected signal waveforms originating from discharge activity (**a**); the visualization of detected discharge activity in the localization module (**b**).

**Figure 11 sensors-19-00923-f011:**
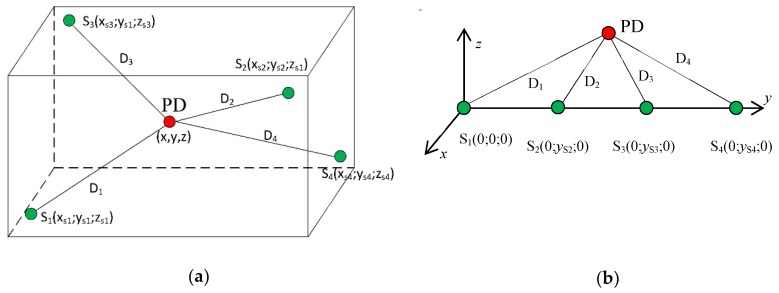
The general (**a**) and linear (**b**) geometrical configurations of the signal source (PD) and sensors S_1_, S_2_, S_3_, S_4_.

**Figure 12 sensors-19-00923-f012:**
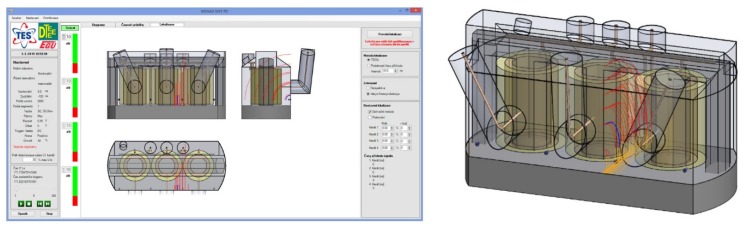
The localization results yielded by the analytical method.

**Figure 13 sensors-19-00923-f013:**
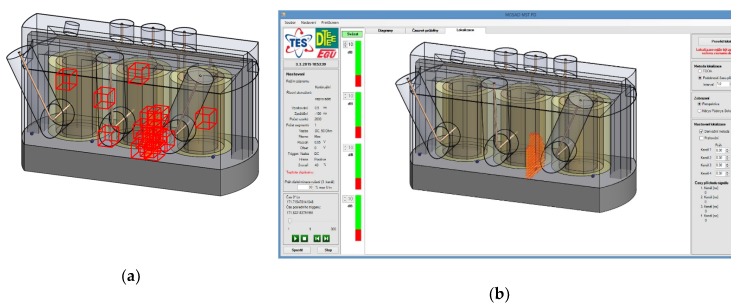
The principle of the comparative method (**a**) and a localization result (**b**).

**Figure 14 sensors-19-00923-f014:**
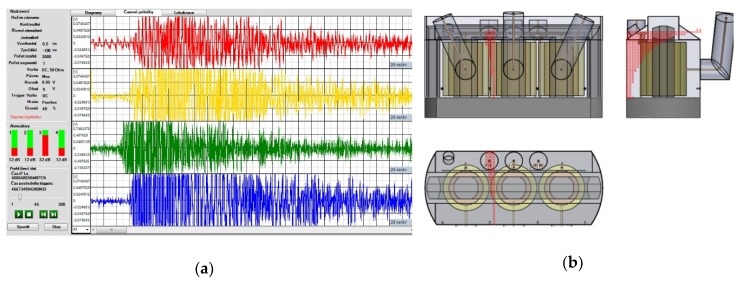
The waveforms of detected signals originating from the discharge activity (**a**); the detected discharge activity in the localization module (**b**).

**Figure 15 sensors-19-00923-f015:**
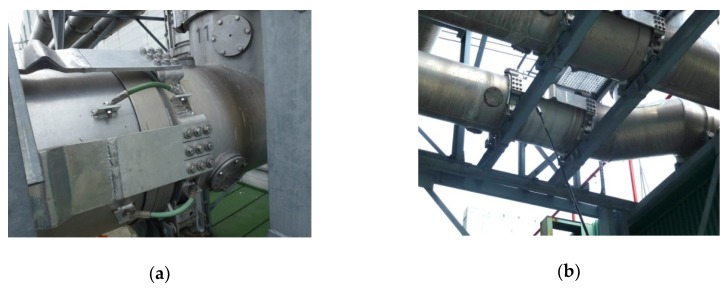
The extendable junction on the feeding bushing for dilatation compensation (**a**); the discharge activity signals close to the bushing dilatation junction examined with a UHF probe (**b**).

**Figure 16 sensors-19-00923-f016:**
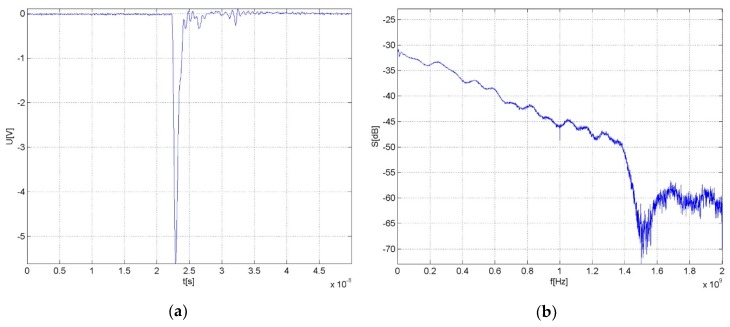
The waveform of the PD simulating impulse signal (**a**); the magnitude spectrum of the simulating signal (**b**).

**Figure 17 sensors-19-00923-f017:**
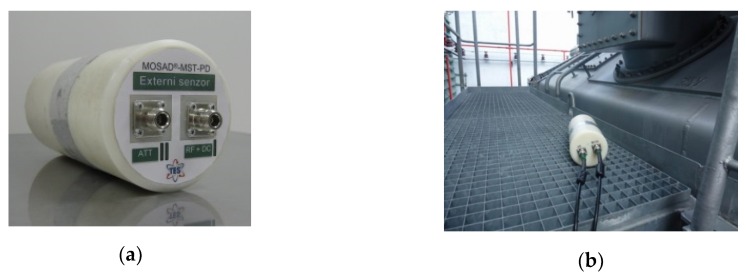
The external sensor for detecting outer EM impulse disturbance (**a**) and its placement on a service footbridge on the top of the transformer during a diagnostic (**b**).

**Figure 18 sensors-19-00923-f018:**
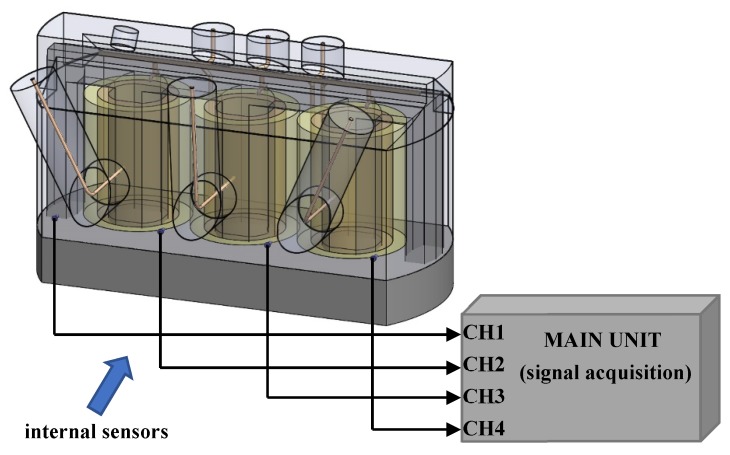
Sensors configuration within the standard detection mode.

**Figure 19 sensors-19-00923-f019:**
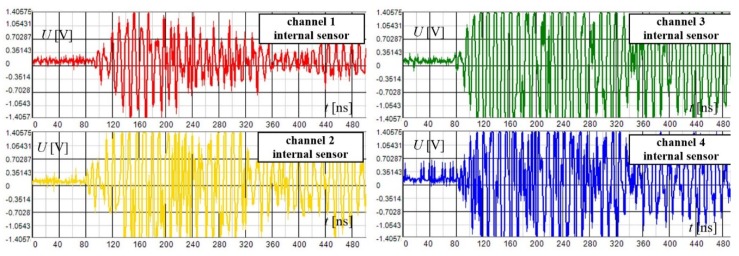
The waveforms of the disturbing signal acquired within the standard detection mode using 4 internal sensors.

**Figure 20 sensors-19-00923-f020:**
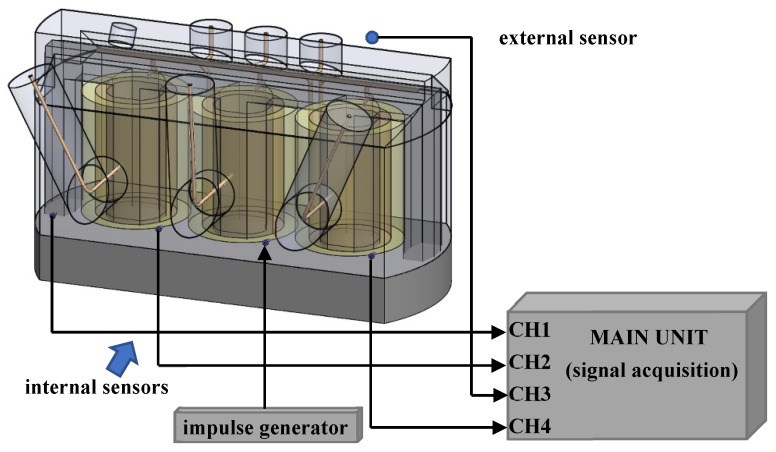
External sensor configuration (utilized within the DDA mode).

**Figure 21 sensors-19-00923-f021:**
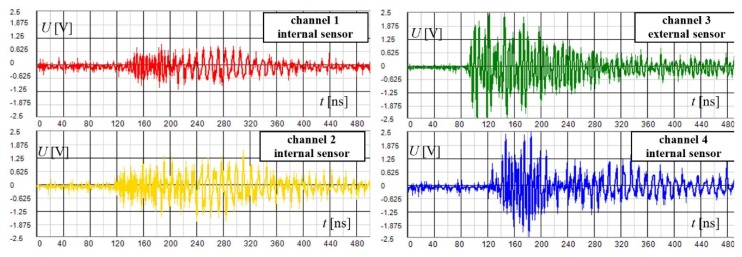
The waveforms of the disturbing signal acquired within the external sensor mode; the external sensor connected to channel 3 (CH3).

**Figure 22 sensors-19-00923-f022:**
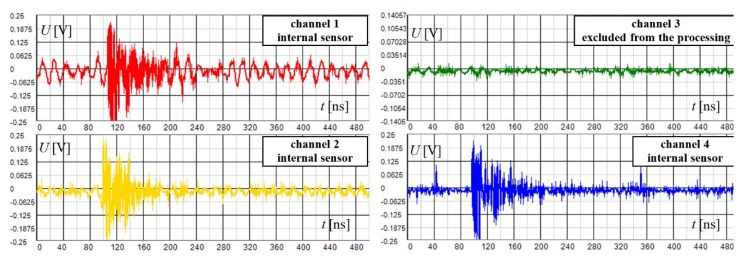
The waveforms of the artificial signal acquired using the DDA method; the signal of the external sensor on channel 3 is excluded from processing and displaying.

**Figure 23 sensors-19-00923-f023:**
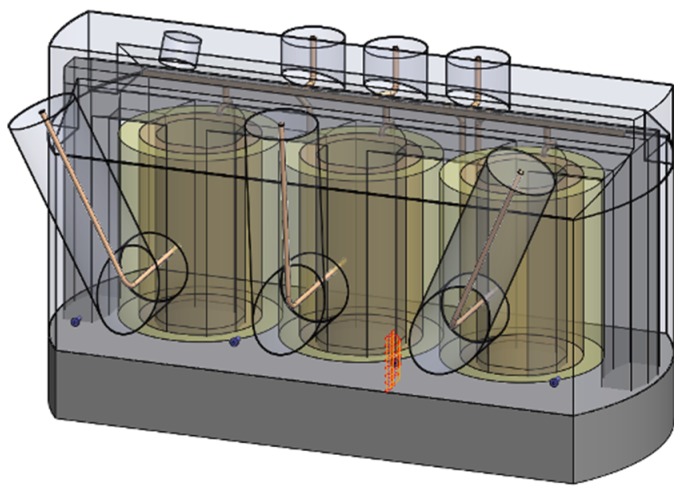
The visualization of the localized artificial signal source position in the third inlet.

**Figure 24 sensors-19-00923-f024:**
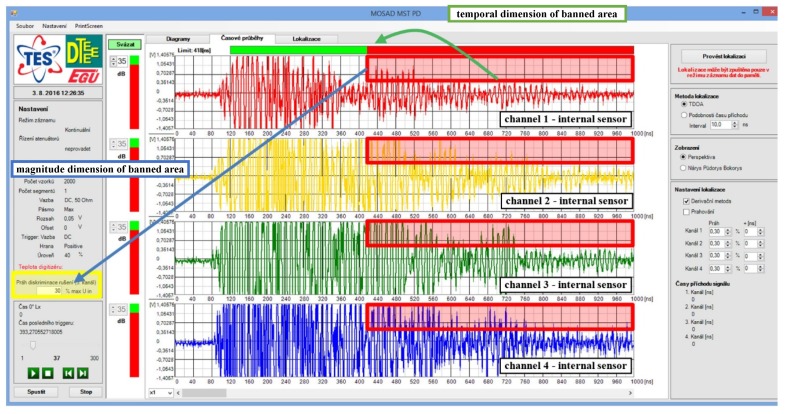
The discrimination based on the signals’ temporal dispersion.

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
