# Peer review of "A Sensor System for Detecting and Localizing Partial Discharges in Power Transformers with Improved Immunity to Interferences"

_sensors, 2019, doi:10.3390/s19040923_

Round 1

Reviewer 1 Report

- Comparative method of "Gating" is not reflected in the discussion or reference list

- What makes you sure, you have a PD in the bushing and not a corona discharge outside the bushing? --> What is the FFT of the signals? --> Corona discharges might be perfectly "shielded" by simple filtering! --> That is a weak point of the full DDA validation and needs to be addressed, otherwise readers might think you haven´t regarded the full picture!

- It is very difficult to follow, when which signals have been injected or measured where? Maybe a picture can help to clarify

- Can you mark in Fig. 18 and followings, what waveform is which channel and what is connected and what is signal source, ...

- Chapter 8 is not really to understand; the discriped "finding" can be seen where? Is that the authors opinion or well proven and accepted in industry? What about the danger of overseeing original waveforms in case the do not fit in the assumptions?

- line 190ff: what is now the bandwidth, the sensitivity of the antenna?

Author Response

Dear reviewer,

we would like to thank you for your proposal to improvements and supplementing our manuscript „Sensor system for detecting and localizing partial discharges in power transformers with improved immunity to interference“.  We have uploaded its revised version.

We have strived to satisfy all of your requirements as far as possible. The changes and supplementation are marked in the manuscript by a blue color. Please let me shortly comment on your remarks:

- Comparative method of "Gating" is not reflected in the discussion or reference list

We agree that "Gating" method was not clearly mentioned in the first version of the manuscript. We have included the description of its principle together with reference on relevant publication, which examines its advantage.

- What makes you sure, you have a PD in the bushing and not a corona discharge outside the bushing? --> What is the FFT of the signals? --> Corona discharges might be perfectly "shielded" by simple filtering! --> That is a weak point of the full DDA validation and needs to be addressed, otherwise readers might think you haven´t regarded the full picture!

We agree that this issue was not clearly emphasized and explained in the manuscript. Our description includes a mention that the sensors contains input high pass filters in order to suppress possible reception of signals with dominant spectral content on lower frequencies up to 100 MHz. The filter is shown in Figure 5 and it is also mentioned in the relevant figure description. We have added hopefully more clear comment, which mentions also the corona issue on lines 216-218 and relevant reference on publication that discuss spectral character of the corona signal.

- It is very difficult to follow, when which signals have been injected or measured where? Maybe a picture can help to clarify.

We have added Figures 18 and 20 to make it more understandable.

- Can you mark in Fig. 18 and followings, what waveform is which channel and what is connected and what is signal source, ...

We have strived to comply your proposal by adding the descriptions.

- Chapter 8 is not really to understand; the discriped "finding" can be seen where? Is that the authors opinion or well proven and accepted in industry? What about the danger of overseeing original waveforms in case the do not fit in the assumptions?

We have improved the text and image content of Chapter 8 to make it better understandable. The forementioned “finding” (temporal waveform dispersion of impulse signal propagating through complex metal-dielectric environment) is in our opinion a known issue in this field. It is discussed in publications that are concerned with PD UHF signal research.

The issue of the danger of overseeing waveforms that do not fit in the assumptions represents the disadvantage of the last discussed approach. To avoid it, the banned area limits must be set manually by a rather experienced operator. We have added a comment on this disadvantage in the Chapter 8.

The manuscript also underwent proposed spell check and connected revision of the English language.

With best regards, on behalf of the team of authors

Petr Drexler

Brno University of Technology, Czech Republic

Reviewer 2 Report

while teh authors effort in developing a new method for PD dtection and localization is valuable

but the paper suffers from lack of comparison between the developed method and the existing methods, It would be better if the authors measure and localise the PD in the considred trasnforemrs and compare the valuse with the results obtainde from their own method, By this 

one can evalute how the developed method is relaible and efficient

Author Response

Dear reviewer,

we would like to thank you for your willingness to go through our manuscript „Sensor system for detecting and localizing partial discharges in power transformers with improved immunity to interference“.  We have now uploaded its revised version. It should more clearly describe the conditions and results of the research.

The manuscript underwent proposed moderate English changes as required.

Regarding your comment on comparison with another method, we would be unfortunately not able to fully comply with your proposal at this time. The focus of the manuscript is to describe the development, implementation and verification of a new detection method that improves the properties of the current PD UHF method. Comparison with the results of another method requires further ongoing research and would, in our opinion, go beyond the extent of the article.

However, we will continue towards this comparison. We will include it in our research plans and we will be working on a following paper on this topic. Thank you for your proposal. I believe that it will significantly help us to evolve the method to became mature for a routine application.

With best regards, on behalf of the team of authors

Petr Drexler

Brno University of Technology, Czech Republic

Round 2

Reviewer 1 Report

The second version is a big improvement against the first one!

Congratulations to the paper!

Author Response

Thank you.

Sincerely,

Petr Drexler